# Intraluminal infusion of Penta-Galloyl Glucose reduces abdominal aortic aneurysm development in the elastase rat model

**Asbjørn Sune Schack** [1,2] *, **Jane Stubbe** [1,3], **Lasse Bach Steffensen** [1], **Hend Mahmoud** [3], **Malene Skaarup Laursen** [1,2], **Jes Sanddal Lindholt** [1,2]

1 Centre for Individualized Medicine in Arterial Diseases, Odense University Hospital, Odense, Denmark,
2 Department of Cardiothoracic and Vascular Surgery, Odense University Hospital, Odense, Denmark,
3 Department of Cardiovascular and Renal Research, Odense University Hospital, Odense, Denmark

* a_schack@hotmail.com

## Abstract

### Background

An abdominal aortic aneurysm (AAA) is a progressive chronic dilatation of the abdominal aorta with terminally rupture when the aortic wall is so weakened that aortic wall stress exceeds wall strength. No effective medical treatment exists so far. We aimed to test whether intraluminal admission of Penta-Galloyl Glucose (PGG) treatment in a rodent AAA model could hold the potential to inhibit aneurysmal progression.

### Method

Male *Sprague Dawley* rats had either intraluminal elastase infused for AAA induction or saline to serve as controls. In two independent experimental series, elastase was used to induce AAA followed by an intraluminal PGG (directly or by a drug eluting balloon) treatment. All rats were followed for 28 days and euthanized. In both series, maximal infrarenal aortic diameter was measured at baseline and at termination as a measure of AAA size. In series 2, maximal internally AAA diameter was followed by ultrasound weekly. AAA tissues were analyzed for elastin integrity by millers stain, collagen deposition by masson trichrome staining. In other AAA tissue samples the mRNA level of CD45, lysyloxidase (LOX), lysyloxidase like protein 1 (LOXL1) were determined by qPCR.

### Results

Direct administration of PGG significantly reduced AAA expansion when compared to controls. PGG treatment resulted in a higher number and more preserved elastic fibers in the aneurysmal wall, while no significant difference was seen in the levels of CD45 and LOX mRNA levels. The drug eluting balloon (DEB) experiment showed no significant difference in AAA size observed neither macroscopically nor ultrasonically. Also the aneurysmal mRNA levels of CD45, LOX and LOXL1 were unchanged between groups.

**Data Availability Statement:** All data nessesary to replicate the findings are available from Data

Archiving and Networked Services (DANS) with
DOI: 10.17026/dans-zq2-ykhg.

**Funding:** Funded by Centre for Individualized
Medicine in Arterial Diseases, Odense University
Hospital.

## Conclusion

A significant reduced expansion of AAAs was observed in the PGG group, suggesting PGG as a drug to inhibit aneurysmal progression, while administration through a DEB did not show a promising new way of administration.

## Introduction

Abdominal aortic aneurysms (AAA) are categorized as a localized diameter enlargement of the abdominal aorta by 50% or more. AAA is most commonly found in men >50 with prevalence estimated between 2–4% of men in the age of 50–60. As several screening programs for AAA have established most of these (85%) are small aneurysms[1], but as age increases the prevalence of bigger aneurysms (>6 cm) increases too[2, 3]. The annual risk of rupture for aneurysms <5.5cm is 1% or lower, while the risk increases significantly as the aneurysm progresses. If an AAA does rupture the mortality is 85–90%[4]. Current treatment options rely on careful surveillance as the aneurysm expands non-linearly and surgical procedures are offered when a threshold is reached. These procedures are both costly and not without risk. If a medical treatment was available, the time window before the necessity of surgical procedures would be an optimal time for medical intervention. AAA development relies on many factors such as elastin degradation, an inflammatory component and genetics. Potential medical treatments are suggested to affect one or several of these factors[5]. Recently, several studies have suggested Penta-Galloyl Glucose (PGG) as a likely candidate to diminish AAA expansion in both rats[6, 7] and pigs[8]. The delivery of PGG in these studies was either perivascular, intravascular or by PGG filled nanoparticle guided by elastin to the injured AAA wall. PGG is a compound found abundantly in various plants and associated with a wide range of preventive properties e.g. able to reduce macrophage oxidative stress and promote inhibition of some inflammatory markers [9]. PGG has been found to stabilize elastin and efficiently inhibits development of experimentally induced AAA, as one of the key factors of aneurysmal growth is the degeneration of elastin. One of the protective genes LOX has been shown to carry an important protective role of aneurysmal growth, by cross linking elastin and collagen [10]. To measure inflammation and immune response we measured a macrophage marker F4/80 [11].

In this study we first aimed to confirm the previous findings of applying PGG intraluminal after AAA induction by intraluminal elastase infusion will inhibit AAA expansion. We chose this model because this model has many similarities to the human AAA, such as the elastin degeneration and the occurrence of intraluminal thrombi (ILT's)[12]. We then aimed to apply PGG with a new more clinical applicable approach using a drug eluting balloon (DEB) to preserve elastin degradation and thereby suppress the development of AAA.

## Materials and methods

### Ethics statement

This study was carried out on 49 male Sprague-Dawley rats weighting 356g to 493g (408 ± 11.2g). 26 rats used in the direct intraluminal method (9 PGG, 9 control, 7 saline) and 23 in the drug eluting balloon experiment (9 PGG, 9 control, 5 saline). Experiments were in accordance with the National Institutes of Health guidelines for use of animal in research and with the approval of by the Danish Animal Experiments Inspectorate (Protocol nr. 2016-15-0201-01046). The rats were allowed to acclimatize for at least one week prior to surgery. All

rats were housed and had free access to water and standard chow, under constant temperature (20C) and humidity (55%) with a 12 hour light/ dark cycle.

## Animal experiments

The abdominal aortic aneurysms were introduced into the rats in accordance with the previous studies [12, 13]. In brief, the rats were anesthetized by a subcutaneous (SC) injection with a mixture of fentanyl (236 ug/kg), fluanisone (7.5 mg/kg) and midazolam (3.75 mg/kg) and given an initial SC saline injection (1mL per 100g body weight). Through a midline laparotomy from processus Xiphoideus to os pubis, the abdomen was opened to expose the intestines which were hereafter put aside and kept moist in a tissue cloth soaked in physiological saline. The aorta was isolated all the way from the aortic bifurcature to the inferior of the two testicular arteries. All the aortic side branches were ligated temporary (6–0, Teleflex; Black Braided Silk Suture, 4-S) with the cranial and caudal part of the aorta ligated last, causing the aortic blood flow to temporarily stop.

**Direct intraluminal PGG intervention through a catheter on AAA expansion.** Through a small transverse arterectomy made just above the caudal ligature, a catheter (Qusina, Tygon tubing, 0.01 x 0.03 inch) was inserted and the aorta was ligated around the catheter. The Aorta was infused for 30 minutes with either elastase 6U/ml or vehicle (physiological saline) resulting in a 130% expansion of the diameter. Then the cathether was removed and the abdomen and aorta was flushed in physiological saline, followed by insertion of another catheter and the aorta was infused with either PGG (0.6mg/ml, Penta-O-galloyl-b-D-glucose hydrate, Sigma-Aldrich Denmark A/S, G7548, >96% (HPLC)) diluted in a solvent (2% ethanol, 2.5% DMSO in isotonic saline) or the solvent alone for 15 minutes.

**Intraluminal PGG intervention through a drug eluting balloon on AAA expansion.** Isolation of the abdominal aorta was performed as describe above. Through a small transverse arterectomy made just above the caudal ligature, a shortened DEB (Prototype from COOK, Bjaeverskov, Denmark—not commercially available, 2.5mm x 30mm) was inserted and the aorta was ligated around the catheter. As in the first experiment, the aorta was infused with either elastase or vehicle depending on the group, resulting in an expansion of the diameter for 30 minutes. After the 30 minutes the DEB was removed and another DEB was inserted and infused with either PGG diluted in a solvent or the solvent alone for 15 minutes.

At the end of infusion in both experiments, the catheter was removed; aorta was closed with a suture (8–0, Ethicon, ETHILON, Polyamide 6/6, W2808) and flow was restored with the removal of the ligatures. The intestines were put back into the abdomen, the abdomen was closed in two layers (5–0, Ethicon, VICRYL, Polyglactin 910, V391), given a SC saline infusion and the rat was placed back in the cage.

**Postoperative care.** Postoperative the animals were for the first three days given Temgesic in Nutella (0,2mg Temgesic crushed and mixed with 1g Nutella; Dosis: 0,4mg/kg) for pain management. The rats were weighted each week and monitored for a total of 28 days. After the 28 days the abdomen were once more opened for measurements as described below. Hereafter the rats were euthanized with a lethal dose of pentobarbital injected into the heart followed by a systemic saline flush. Both the treated and an untreated-control segment of the infrarenal abdominal aorta were removed and divided into three pieces each. Two of the three pieces were snap-frozen in liquid nitrogen and stored at –80C in order to measure changes in mRNA expression patterns while the last piece was fixed in 10% normal formalinbuffer (Hounisen Laboratorieudstyr A/S, Lot 03800838), and placed in 1x PBS with acid 0.1% azid 24 hours later and then embedded in paraffin for histological analyses.

**Measurements of outer aortic diameter.** Images of the aorta were obtained with a digital camera (Canon 6D) attached to the operating microscope for measurements of both the length of the affected area and the widest aortic diameter. A 10mm scalebar (Micro-Tools, Micro Scale, Tip 0.5mm) were placed in the same plane and next to the aorta in order to calibrate measurements. The "before" image was obtained after the initial isolation of the aorta (Fig 1A); the "under" image was done at the time of infusion of saline/elastase (Fig 1B); the "after" image was done 5 minutes after reestablished blood perfusion (Fig 1C) and a last image was done just before euthanization after the 28 days after dissection of aorta (Fig 1D).

At the time of imaging a measurement was done of both the length of the affected area and the largest diameter of aorta.

**Inner abdominal aortic diameter measurements.** In the experiment with the DEB, a baseline ultrasound measurement (LogiQ e with L10-22-RS transducer) was performed in the horizontal (Anterior-Posterior-measurements, AP) plane before the initial aorta isolation, by an individual of whom the treatment of the rats had been blinded. These measurements were repeated postoperative day 3, 7, 14, 21 and 28. An inner-to-inner (ITI) diameter of the aneurysm was measured in the AP plane after an initial evaluation of the largest diameter in the axial plane, where rat treatment was blinded to the observer.

**Intraobserver variation.** To estimate intraobserver variation with ultrasound, a measurement was done of 10 random rats, which was not yet exposed to any treatment. A measurement of the abdominal aorta was done in both the sagittal and axial plane to evaluate the widest diameter just above the bifurcature.

After the first measurements the rats were then again chosen at random and re-measured in the same way, to evaluate the reproducibility of the measurements. The 95% interquartile for the first axial diameter was (2.04;2.38mm) and the sagittal was (2.07;2.33mm). The re-test showed the axial to be (2.11;2.50mm) and a sagittal of (2.09;2.36mm).

## Histology

Cross sections of 5-μm of the rat AAAs at the widest diameter was deparaffinized, and stained with hematoxylin and eosin for general structures, Millers elastin stain (Atom Scientific, RRSK11) to identify for elastic structure according to manufactures instructions. In addition, sections were stained by Masson trichome (Sigma, HT15) for identification of collagen fibers, fibrin and nuclei.

The elastin fibers were counted as a way to evaluate the damage done to tunica intima and the elastin fibers. The number of elastin fibers was counted in the most northern/southern/eastern and western point of the histological sample. These points were chosen to ensure reproducibility of every sample. The total count of elastin fibers were divided by four to get a mean value of fibers (Fig 2).

**Determination of aneurysmal RNA levels.** Frozen AAA tissue was disrupted by shaking 3 min at 3000 Hz in Tissue Lyser II in 500 ul Trizol solution (Invitrogen, Thermo Fisher) and a 5 mm stain-less steel beads (Qiagen). Then samples were incubated 10 min at room temperature, before adding chloroform for RNA separation. RNA in the upper phase was then precipitated in isopropanol, and was in 70% ethanol before eluting in RNase free water. 1 ug total RNA was treated with DNAse digestion kit (Thermo Scientific) according to manufacturer's instructions. Then cDNA was generated using iScript (biorad) according to manufactures instructions. 50 ng cDNA of each rat AAA sample was used in duplicate to determine mRNA levels of F4/80 (5'– TTTTGGCTGCTCCTCTTCTG –3';5'– TGGCATAAGCTGGACAAGTG –3'), Lysyl oxidase (5'–ACCTGGTACCCGATCCCTAC3'; 5'–AGTCTCTGACATCCGCCC TA–3'), Lysyl oxidase like protein (5'– ACTTGCCTGTGCGAAACTCT –3'; 5'–CCTGCA

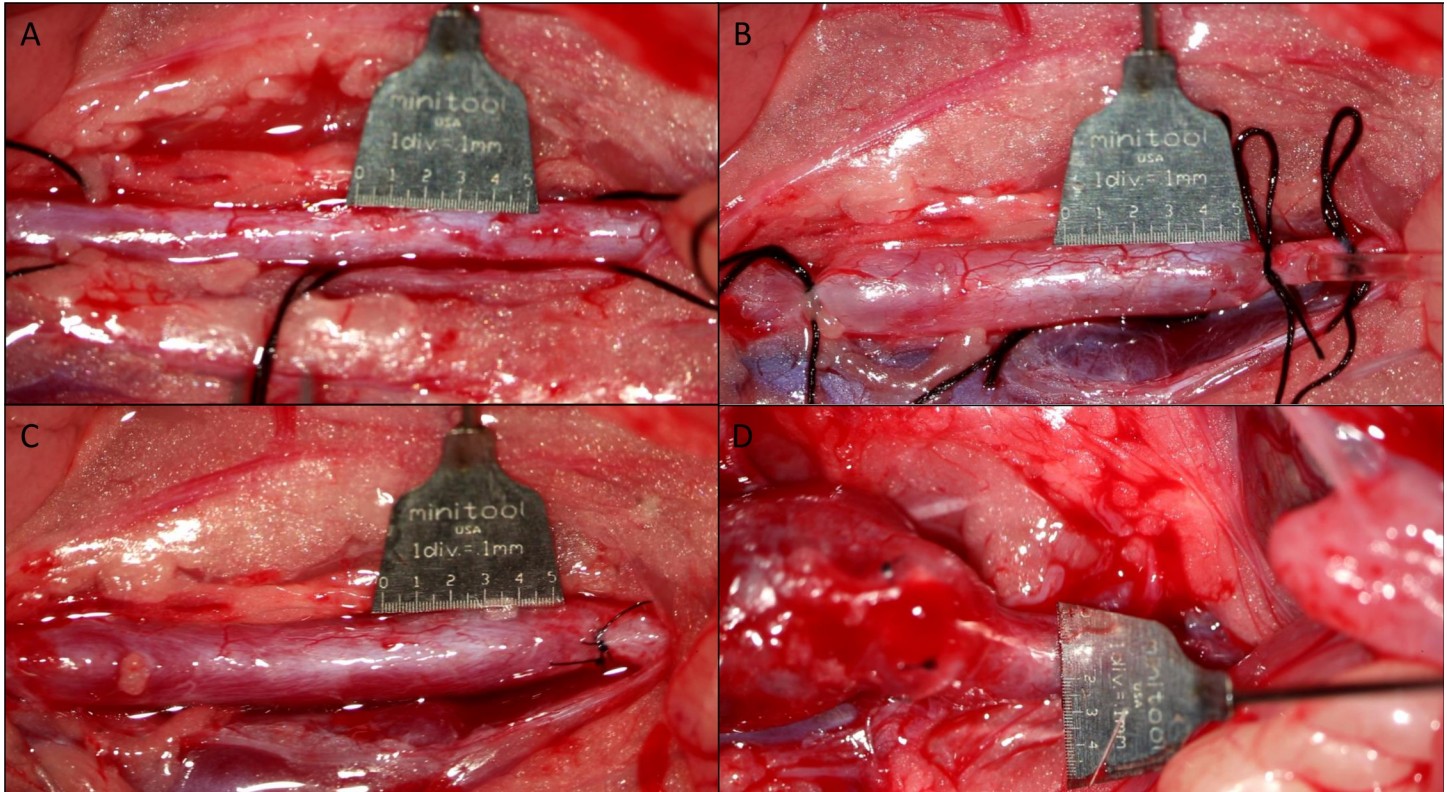

**Fig 1.** (A) Infrarenal aorta before ligation and infusion, (B) Infrarenal aorta after ligation and under infusion, (C) Infrarenal aorta after removal of ligations and catheter and showing reestablished bloodflow, (D) Infrarenal aorta at the time of euthanization 28 days later, showing an aneurysm.

CGTAGTTGGGATCT-3'), and the house keeping gene TBP (5' - CAGCCTTCCACCTTATG CTC -3'; 5'-TGCTGCTGTCTTTGTTGCTC -3'), by quantitative RT-PCR using SYBR-green (Biorad) as detector in a 3 steps PCR protocol (40 cycles: 95˚C 20; 60˚C 20 sec; 72˚C 20 sec). For each PCR product of interest, a standard curve was constructed by plotting threshold cycle against serial dilutions in order to determine efficacy and linearity. Furthermore a melting curve was run at the end of each qRT-PCR to confirm selectivity. As negative control we used water and RNA samples (without reverse transcriptase).

## Statistics

Normal distribution of both weight progression and before/after aortic diameter data was tested by Kolmogorov-Smirnov tests.

Grubbs' test was used to determine whether the most extreme value of the results were a significant outlier. A single outlier in the PGG group of the DEB experiment was identified and excluded in further analyses.

Paired students t-tests were used to compare results between the two groups in the directly PPG administrated experiment. In the DEB experiments a repeated measure multivariate analysis of variance (MANOVA) was used to determine the correlation of the aortic diameters between the groups over time, while a one-way analysis of variance (ANOVA) was used when comparing outcome and starting value.

RNA levels were analyzed by one way ANOVA or student's t-test. Results are presented as means ± standard derivation (SD). P-values <0.05 were considered significant.

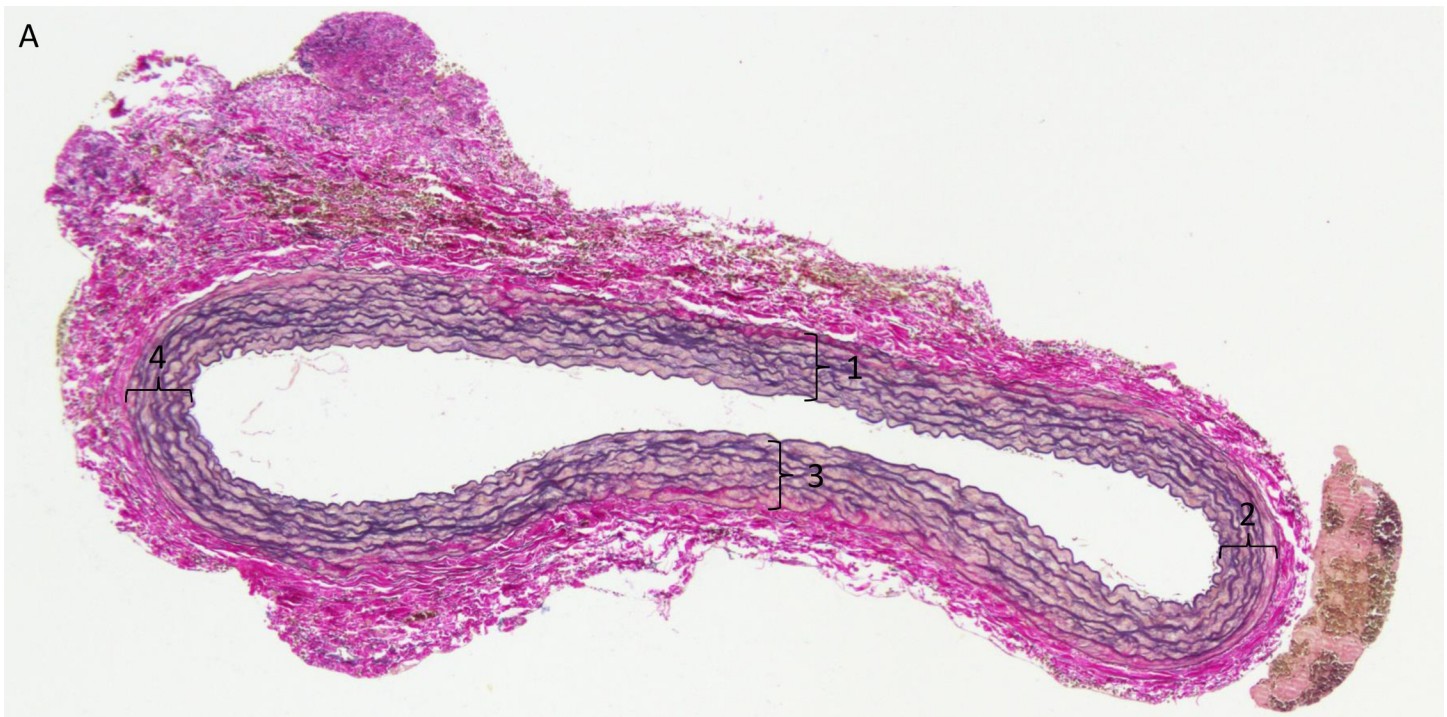

**Fig 2.** (A) The four points of measurement for elastic fibers (1–4).

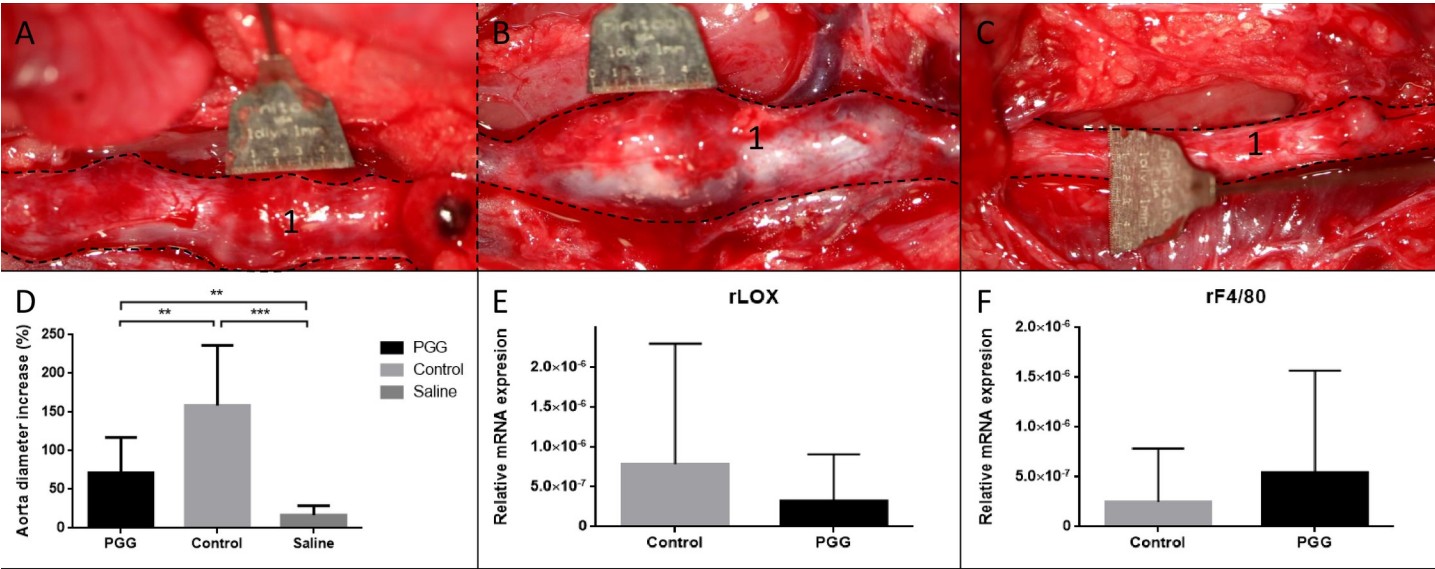

**Fig 3. Results of the direct PGG administrated experiment.** Perioperative picture of the infrarenal aorta (1) 28 days after primary surgery in the PGG treated group (A), the control/ elastase treated group (B) and the saline treated group (C). PGG (n = 9) resulted in development of smaller AAA based on maximal aortic diameter when compared to the elastase treated group (n = 10, p<0.01). Both the PGG treated and the saline treated group had larger aortic diameters when compared to the saline treated controls (n = 7, p<0.01 and p<0.001, respectively), graphically presented in (D). Quantitative mRNA levels of LOX (E) and F4/80 (F) showed no significant differences in mRNA expressions, p = 0.40 and p = 0.43, respectively. Both quantification levels were normalized to tatabox binding protein (TBP). Data are presented as mean ± SD. (*) indicates p<0.05, (**) indicates p<0.01 and (***) indicates p<0.001.

## Results

### Direct intraluminal PGG intervention through a catheter on AAA expansion

In order to determine if directly intraluminal PGG intervention could prevent AAA expansion rats were divided by body weight (mean starting weight 408 ± 29.4g) in 3 groups; elastase followed by PGG (PGG treated group); elastase followed by saline (control group) and a group given saline followed by solvent (saline) to determine the effect of mechanical stress on the aortic wall. Over the experimental period (28 days), there were no difference in body weight between the groups (n = 9, n = 10, n = 7; p = 0.17). They all seemed to thrive and gained 59.1 ± 11.2g in the PGG group, 52.4 ± 10.9g in the control group and 57.2 ± 9.48g in the saline group, with no significant difference between the groups (p = 0.1). Also organ to bodyweight did not change, with heart to bodyweight: (PGG 0.33 ± 0.03% control 0.32 ± 0.03% and saline 0.33 ± 0.03%) with p = 0.30 and kidney to bodyweight: (PGG 0.54 ± 0.06%, control 0.54 ± 0.05% and saline 0.52 ± 0.01%) with p = 0.24.

**AAA development.** The starting mean outer abdominal aortic diameter were 2.01 ± 0.13mm and was also comparable between the groups (PGG 2.04 ± 0.16mm; control 2.02 ± 0.12mm and saline 1.96 ± 0.1mm) with p = 0.42. After 28 days after AAA induction both elastase treated groups had developed AAA (Fig 3), determined by maximal outer abdominal aortic diameter (PGG increased 71.4 ± 46% and controls increased 159 ± 77.5% increase), while the saline group did not show any signs of an abdominal aortic diameter with just a small increase (17.0 ± 11.4%). The terminal aneurysm size in the PGG treated group was significantly smaller than in the elastase treated group (Fig 3, PGG and Control; 3.48 ± 0.91mm and 5.24 ± 1.61mm, p< 0.01).

**AAA composition.** In general the PGG treated group had a tunica media with regularly arranged parallel elastic fibers with only a few examples of ruptured elastic fibers and a thin layer of intraluminal thrombus with an otherwise intact intima and medial layer (Fig 4B and 4E). No clusters of infiltrating leucocytes or an augmented neovascularization of the adventitial layer was observed and it seemed largely unaffected.

This is in contrast to the elastase treated (control) group, that typically had a larger intraluminal thrombus and an extensive defragmentation or complete disruption of the elastica interna (Fig 4C and 4F). Furthermore, a widened intercellular space was shown along with defragmentation of both the elastic and collagen fibers (Fig 4C). In addition there were no clear separation between the medial and adventitial layers while also showing signs of infiltrating leucocytes and neovascularization (Fig 4F). In the saline treated group, all samples had a normal looking aortic wall composition with a well preserved intima, media and adventitia layer. There were no observed fragmentation of the elastic fibers or collagen fibers and clear separation of the layers was easy to identify (Fig 4A and 4D).

As a measure of elastic integrity the saline group displayed a mean value of elastic fibers (EF) of 9.25 ± 1.66 EF, n = 7. This were similar in the PGG group (9.3 ± 2.24 EF, n = 9, p = 0.47), while the controls had significantly more degraded elastic fibers when compared to PGG and saline treated group (6.9 ± 1.94 EF, n = 10, p<0.01).

Changes in mRNA levels in the aneurysmal tissue revealed no differences in aneurysmal mRNA levels of the macrophage marker F4/80 between the PGG treated group and elastase control rats (Fig 3F). As an indicator of elastogenesis, aneurysmal LOX mRNA levels were measured. Here the highest mean level was measured in the elastase treated group when compared to PGG treated group, though non-significantly (Fig 3E).

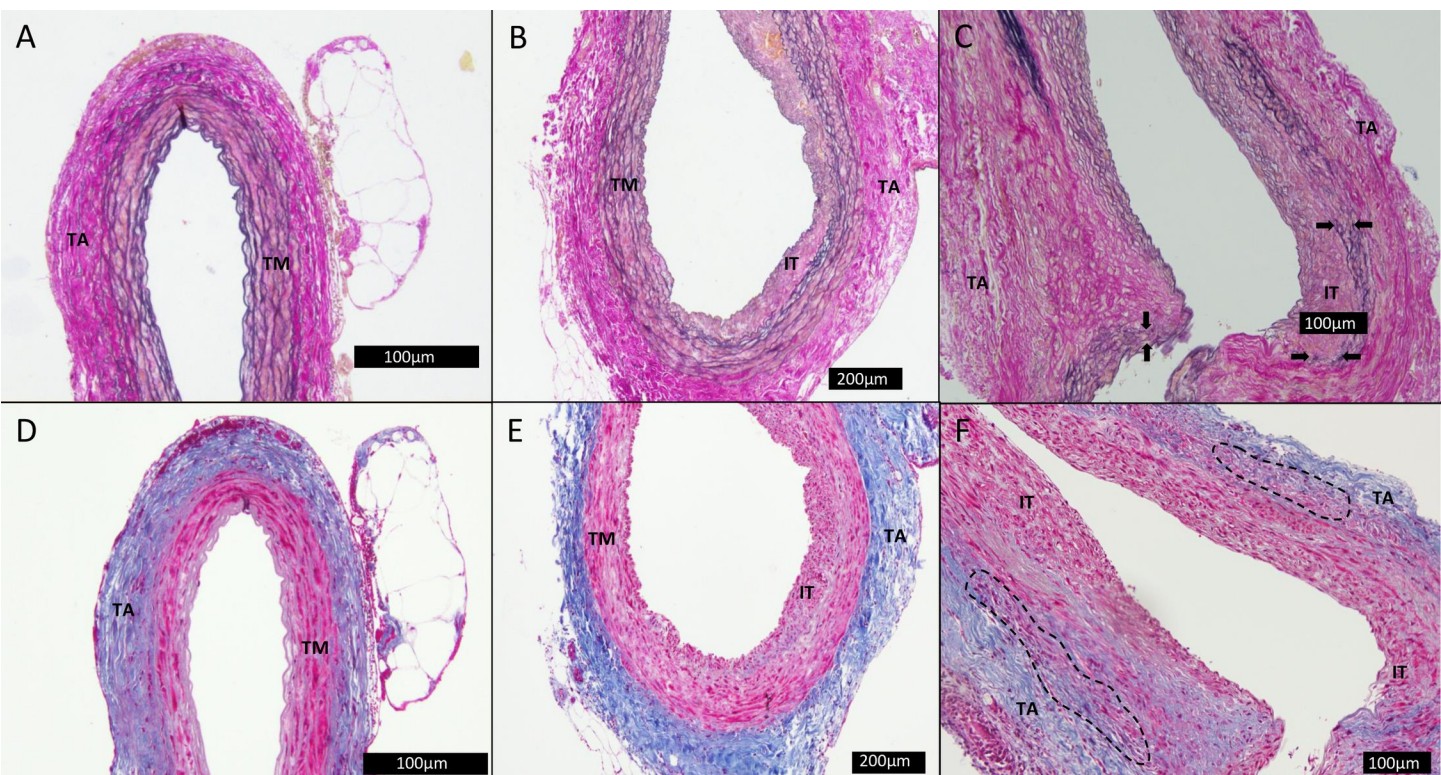

**Fig 4. Histological slices of the abdominal aorta.** The slices, A-C, were stained with Millers elastin stain, while the slices, D-F, were stained with Masson trichrome. Slice A and D were neighboring slices from same rat in the saline group, slice B/E were from the PGG group and C/F from the control group. TA: tunica adventitia. TM: tunica media. IT: intraluminal thrombus. Black arrows: marking of defragmentation. Dashed line: example of area with neovascularization and infiltrating leucocytes.

### Intraluminal PGG intervention through a drug eluting balloon on AAA expansion

With the same procedure as described above the rats were divided by body weight (mean starting weight 461 ± 29.2g) in 3 groups PGG group (elastase followed by PGG), control group (elastase followed by solvent) and saline group (saline followed by solvent). This time elastase and PGG and saline were delivered through a more suitable clinical setting by a DEB. Two rats died (one in the PGG treated group and one in the elastase treated control group) of an abdominal aortic dissection/rupture, which was determined at autopsy before reaching experimental day 14 and these rats were not included in any of the analyses described below. The PGG group gained 107 ± 31.7g, the controls gained 98.2 ± 26.5g and the saline group gained 120 ± 26.36g, with no significant difference between the groups (p = 0.11).

**AAA development.**   Mean starting outer abdominal aortic diameter measured was 2.14 ± 0.13mm and with mean inner abdominal aortic diameter determined by ultrasound was 2.13 ± 0.13mm.

No expansion of the abdominal aorta was observed with ultrasound in any of the three groups at day 3. The first abdominal aortic expansion was seen at day 7 in both the PGG- and elastase treated control group, while the saline treated group did not show any signs of abdominal aortic expansion (Fig 5).

At day 28 both the PGG and elastase control treated group had developed AAA, while the saline group did not show any signs of aneurysm formation based on both ITI aortic diameter and outer abdominal aortic diameter (Fig 5). In all groups the increase of the widest outer

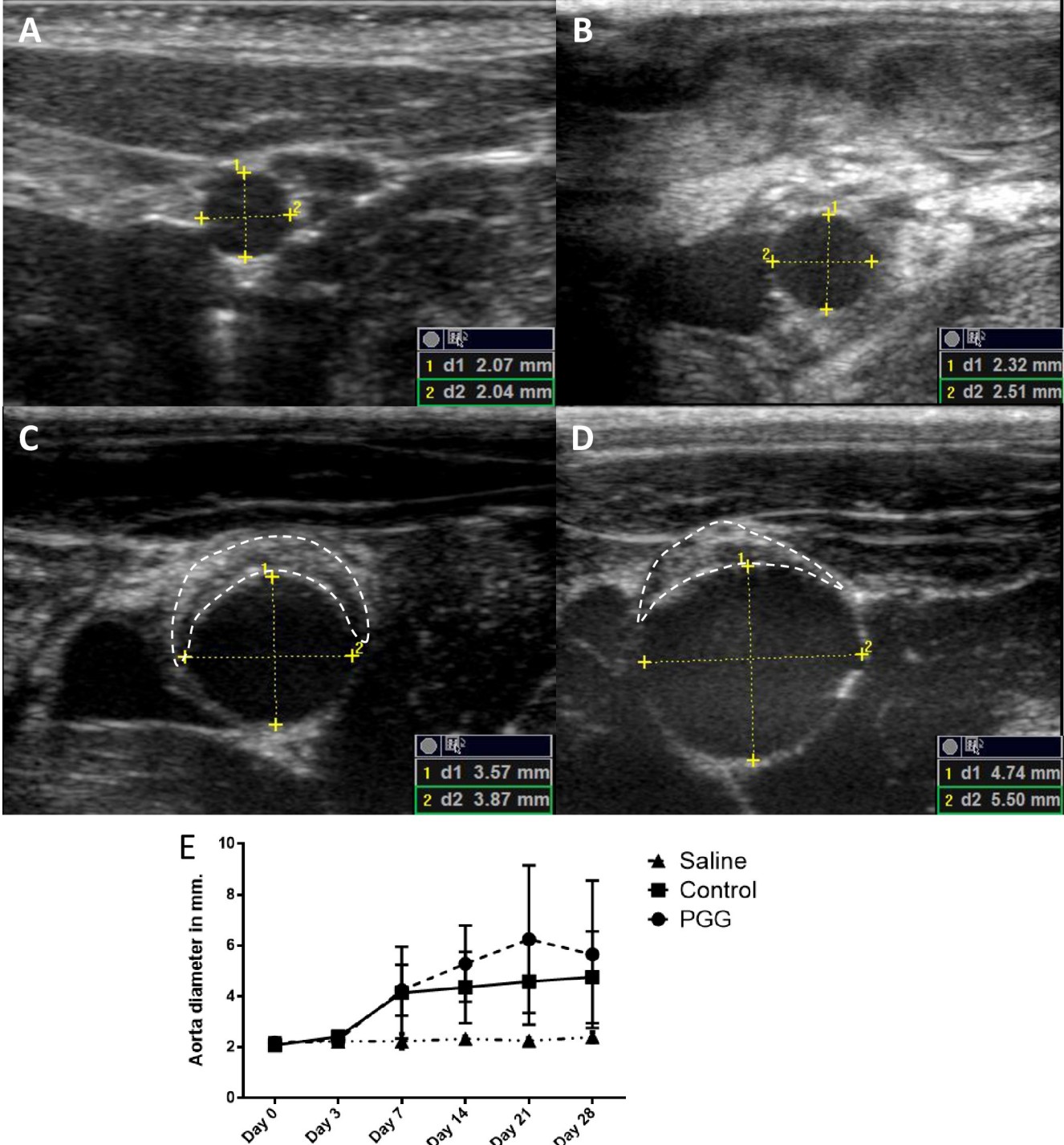

**Fig 5.** Ultrasonic pictures of the infrarenal aorta of a rat in the elastase treated group at day 0 (A), day 3 (B), day 7 (C) and day 14 (D). The clear development of dilatation of the infrarenal aorta was seen, as well as intraluminal thrombus formation (the area within the dashed white line) from day 7. The yellow markings showed the points of measuring both dilatation in AP plane (1) as well as horizontal plane (2). The schematic development of dilatation in the infrarenal aorta was visualized in both the saline (triangle), control (square) and the PGG (circle) treated group, though no dilatations were seen in the saline treated group (E). Data are presented as mean ± SD.

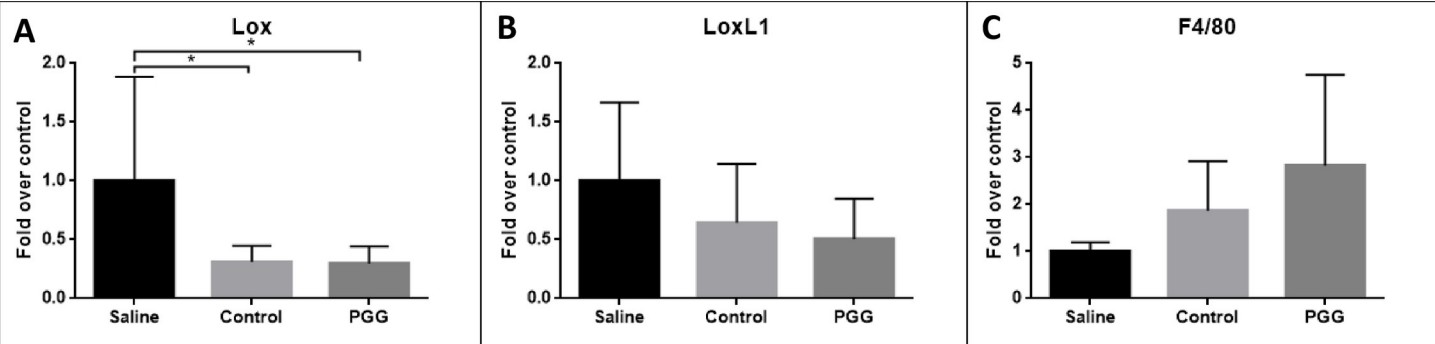

**Fig 6.** Quantitative mRNA levels of LOX (A), LOXL1 (B) and F4/80 (C). There were a significant larger expression of LOX in both the control group and the PGG treated group when comparing to the saline treated group, p = 0.03 and p = 0.02. There were no significant differences in expressions of LOX1 and F4/80. All quantification levels were normalized to tatabox binding protein (TBP). Data are presented as mean ± SD. (*) indicates p<0.05.

abdominal aortic diameter and the widest inner abdominal at day 28 was evaluated. In the saline treated group the increase in outer abdominal aortic diameter increase (12.6 ± 7.39%) and inner abdominal aortic diameter (13.3 ± 13.9%) was comparable. While both the PGG treated group and the control group had a significant increase in abdominal aortic expansion, there was however a discrepancy between the increase in outer abdominal aortic diameter (183 ± 59.1%; 149 ± 104%, respectively) and inner abdominal aortic diameter (143 ± 91.5%; 129.2 ± 97.3%, respectively).

To our surprise, applying PGG via a DEB did not significantly lower neither the outer (PGG: 6.13 ± 1.01mm; control: 5.15 ± 1.96mm, p = 0.46, n = 15) nor inner abdominal aortic diameter (PGG: 5.22 ± 1.06mm; control: 4.75 ± 1.91mm, p = 0.57, n = 15) when compared to the elastase threated control group. We next examined if there were a difference in the incidence of intraluminal thrombi after PGG treatment.

**Intramural thrombus.** Already after 7 days, intraluminal thrombi's (ILT) were observed (Fig 5C and 5D), in 7/9 rats in the PGG treated group and 5/9 in the elastase treated control group, while none was found in the saline treated group. After 14 days only 3/9 in the PGG treated group and 2/9 in the elastase treated control group remained to be seen and these remained until the end of the experiment at day 28.

**Histology and mRNA analyses.** Again the aneurysmal mRNA levels of F4/80 was determined as an indicator of infiltration of macrophages, the mean F4/80 mRNA levels were non-significantly increased in the PGG treated and elastase treated control group when compared to the saline treated group (Fig 6). As an indicator of elastin generation and organization aneurysmal mRNA levels of LOX and lysyloxidase like protein 1 (LOXL1) were measured. Again no differences in relative mRNA levels were detected in any of the transcript measured between the groups (Fig 6). However there were a strong trend towards decreased levels of LOX in both the PGG and the elastase treated group when compared to saline controls.

## Discussion

In this study the group given PGG showed a more normalized histological structure, while also proven a stabilizing effect on the abdominal aortic aneurysm expansion in rats when provided intraluminally to the aorta. This stabilizing effect failed when attempted to administrate it with a DEB, which would be more relevant for clinical adaption. The stabilizing findings are in line with the results from Kloster et al[8], using a porcine model and what Isenburg et al[6] and Nosoudi et al[7] showed in their CaCl$_2$ induced AAA rat model. We based our studies on

the elastase induced AAA in rats as it has many similarities to human AAA development, in regards to the elastin degradation, macrophage infiltration and thrombus formation, while these are not developed consistently in either the murine, porcine models or the CaCl$_2$ rat model [12]. Besides the elastin stabilizing effect, others have found that PGG also modulate immune reactions [14, 15], which also play an essential role in the expansion and rupture of AAAs [5, 16]. However, based on our aneurysmal mRNA levels of F4/80 we do not see any effect of PGG on infiltrating macrophages.

However, even though there were no differences in the levels of F4/80 mRNA levels we cannot rule out that a shift in the balance between M1 and M2 towards M2 inflammatory and tissue regenerating macrophages has occurred as described by Dale et al [17] and could explain the protective effects against AAA expansion in the PGG treated group.

An increased elastogenesis could also explain the protective effects of PGG on AAA expansion. LOX and LOXL1 have been found to be involved in the stabilizing mechanisms of the aortic wall under the progression of AAA [18]. LOX and its related proteins are critical for the crosslinking mechanisms of structural proteins such as collagen and elastin that preserve vessel wall integrity [10]. We did however not find increased levels of aneurysmal LOX and LOXL1 mRNA levels from our PGG treated rats as seen by Nosoudi [7]. We even detected a non-significant trend toward lower mRNA levels of LOX and LOXL1 in the aneurysmal wall when compared to the saline treated rats. Our measurements were done at mRNA levels and therefore may not necessary correlate with either protein levels or protein activity which was determined by Nosoudi et al [7].

The morphological evaluation of the aneurysms in the PGG group led to a larger number of preserved elastic fibers compared to the control group. This is in line with previous findings [13] also showing a protection of the elastic fibers by PGG treatment. In addition, both Isenburg et al [6] and Nosoudi et al [7] showed a more preserved elastic integrity determined by total desmosine content in their studies. Also in the porcine AAA model by Kloster et al [8] elastin seemed to be preserved. Thereby indicating this protective effect of PGG is not species specific.

It is believed that PGG protects the elastin fibers from degradation by attaching to the fibers and preventing proteolytic enzymes to reach their cleavage sides. Other studies have detected PGG associated with the elastic fibers using FeCl$_3$ [6, 7], that leads to an oxidation of Fe when FeCl3 reacts with PGG. Despite several attempts using different methods to visualize the PGG (both FeCl$_3$ and a 4',6-diamidino-2-phenylindol (DAPI) stain, a phenol-binding stain visible in ultraviolet lighting [19] we were unsuccessful. It is noticeable that only in studies using the CaCl$_2$ model in rats, the visualization of PGG using FeCl$_3$ has been successful. We do however believe that PGG is attached to the elastic fibers in our model as well.

In our second experimental approach PGG was applied using a DEB. The DEB is a new and innovative method to administer PGG to a targeted location in an animal model. This approach was included in the hopes of providing the clinicians the tool to prevent continuous growth of small AAA, and type IA+B endoleaks after abdominal endovascular aneurysm repair (EVAR) by placing PGG in the aortic sealing zones by DEB. However, we couldn´t demonstrate impairment of the aneurysmal progression. That obviously raises the question why not? We can only speculate but several explanations seem to exist:

1. Failed measurement methods. In the DEB experiments, AAA expansion was determined by inner aortic diameter measured ultrasonically. There were a good correlation between the maximal outer aortic diameter and inner aortic diameter at the time of surgery, as no peri-adventitial adhesions (fibrous scar tissues) or intra luminal thrombi had yet occurred. However at the end of the experiment, the correlation between inner and outer abdominal aortic diameter was lost, these findings have been detected by others [20]. Both intra-abdominal

adhesions and variations of ILTs after the surgery complicated the correlation of outer and inner measurements. However, we also used the same method of measurement as in the directly administrated experiment. Neither showed a difference suggesting that the lack of effect isn´t due to information bias caused by measurement methods.

When examining the aneurysmal mRNA levels of F4/80, LOX and LOXL1, no differences were detected between the groups. This is in contrast to the directly PGG administrated experiments, which showed significantly better preservation of the elastic fibers in the aortic medial tunic in the PGG treated rats. It causes suspicion on the delivery system:

2. It didn´t pass through the balloon as planned, but tested macroscopically, it did.

3. It didn´t pass through into the wall. The directly administration which worked was applied through some pressure, perhaps the DEB didn´t deliver it with a similar needed pressure. However, the DEB was originally designed for human coronary arteries with an inflated balloon size of 2.5mm and a length of 30mm. This did not fit perfectly for our model and the balloons had to be modified by both shortening and ligation. This procedure could possibly have damaged the balloon and therefore had a harmful effect on the intraluminal arterial wall.

4. The used dose was insufficient. It was based upon the experience from the first series, but that could be insufficient for application this way.

5. PGG simply doesn´t work. This seems unlikely due to the positive effect seen in series 1 and existing literature.

In conclusion, the failure of the DEB administration in the current experiment is likely to have been caused by the delivery system or insufficient dose. However, it is currently impossible to clarify that further. Nevertheless, an intraluminal method to stabilize AAA and aneurysm-prone aortic segment does still seem to be a promising approach for patients diagnosed with an early stage of AAA, while nanoparticle delivery also seems to be a promising approach it is still a novel experimental with unknown long term organ effects. Future experiments need to focus on and fix these problems, document a similar benefit as seen by direct application before translation into humans. It could be very interesting to see the use of a DEB balloon used on larger model, e.g. a porcine model, as this would have not only the size and closer anatomical resemblance to humans, but also because it would be possible to use already tested and used DEB because of the size.

## Acknowledgments

We would like to thank our skillful technicians Lene Bundgaard Andersen, Kenneth Andersen, Amalie K. Mogensen and Inger Nissen.

## Author Contributions

**Conceptualization:** Asbjørn Sune Schack, Lasse Bach Steffensen, Jes Sanddal Lindholt.

**Data curation:** Asbjørn Sune Schack, Lasse Bach Steffensen, Hend Mahmoud.

**Formal analysis:** Asbjørn Sune Schack, Jane Stubbe, Lasse Bach Steffensen, Jes Sanddal Lindholt.

**Investigation:** Asbjørn Sune Schack.

**Methodology:** Asbjørn Sune Schack, Jane Stubbe, Lasse Bach Steffensen, Jes Sanddal Lindholt.

**Resources:** Asbjørn Sune Schack, Jane Stubbe, Jes Sanddal Lindholt.

**Software:** Asbjørn Sune Schack, Jane Stubbe, Lasse Bach Steffensen, Hend Mahmoud.

**Supervision:** Jane Stubbe, Lasse Bach Steffensen, Jes Sanddal Lindholt.

**Visualization:** Malene Skaarup Laursen.

**Writing – original draft:** Asbjørn Sune Schack.

**Writing – review & editing:** Asbjørn Sune Schack, Jane Stubbe, Lasse Bach Steffensen, Malene Skaarup Laursen, Jes Sanddal Lindholt.

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
