## [Decision Letter · Decision Letter 0]

2 Apr 2020

PONE-D-20-06721

Intraluminal Infusion of Penta-Galloyl Glucose Reduces Abdominal Aortic Aneurysm Development in the Porcine Pancreatic Elastase Rat Model

PLOS ONE

Dear dr Schack,

Thank you for submitting your manuscript to PLOS ONE. After careful consideration, we feel that it has merit but does not fully meet PLOS ONE’s publication criteria as it currently stands. Therefore, we invite you to submit a revised version of the manuscript that addresses the points raised during the review process.

We would appreciate receiving your revised manuscript by May 17 2020 11:59PM. To enhance the reproducibility of your results, we recommend that if applicable you deposit your laboratory protocols in protocols.io, where a protocol can be assigned its own identifier (DOI) such that it can be cited independently in the future. For instructions see: http://journals.plos.org/plosone/s/submission-guidelines#loc-laboratory-protocols

We look forward to receiving your revised manuscript.

Kind regards,

Michael Bader

Academic Editor

PLOS ONE

Journal Requirements:

2. In your Methods section, please provide additional information on the total number of animals used , and on how they were allocated to different experimental groups.

3. Thank you for stating the following in the Sources of Funding Section of your manuscript:

"Funded by Centre for Individualized Medicine in Arterial Diseases, Odense University Hospital"

"The author(s) received no specific funding for this work.

Reviewers' comments:

Reviewer's Responses to Questions

**Comments to the Author**

1. Is the manuscript technically sound, and do the data support the conclusions?

Reviewer #1: Yes

Reviewer #2: Yes

2. Has the statistical analysis been performed appropriately and rigorously? 

Reviewer #1: Yes

Reviewer #2: Yes

3. Have the authors made all data underlying the findings in their manuscript fully available?

Reviewer #1: Yes

Reviewer #2: Yes

4. Is the manuscript presented in an intelligible fashion and written in standard English?

Reviewer #1: Yes

Reviewer #2: Yes

5. Review Comments to the Author

Reviewer #1: The study by Asbjørn sune Schack et al. addresses the role of penta-galloyl glucose (PGG) in the aneurysm formation. In an elastase induced model of abdominal aortic aneurysm (AAA) in the rat, direct administration of PGG significantly reduced aneurysm expansion and preserved aortic structure compared to controls. Nevertheless, administration of PGG via a drug eluting balloon was not effective. The study confirms previous investigations on the role of PGG in preventing of AAA and describes different administration form of the drug. This study contributes to our knowledge about AAA treatment but several limitations exist:

Comments:

1. If possible, please provide evidence that penta-galloyl glucose does not precipitate with elastase.

2. How the decision of PGG concentration was made?

3. What was the reason of 28 days study? The elastase induced AAA is known

to develop in 1-2 weeks.

4. The regulation of LOX was studied in the extracts derived from the whole aorta samples. The structure of the aorta, the ratio media /adventitia was dramatically changed in 28 days after AAA induction as shown by histology. Therefore, the regulation of LOX and CD45 mRNA is difficult to interpret. Could you provide immunohistological data about LOX Regulation in PGG Group?

6. The innovation of this study should be more exactly formulated in the discussion.

7. In Figure 5 (2) the graph is not understandable: Which one is PGG group and which is control?

Reviewer #2: The goal of the study was to evaluate the ability of intraluminally infused PGG to influence the outcome of AAA in the elastase AAA rat model. Overall the paper is not impressive. The concept is not new as others have shown that PGG binds to aneurysmal elastin and prevents its degradation, slowing down AAA progression. The first study, with PGG infusion is not entirely novel. The second study, albeit more novel, shows lack of success of a PGG-loaded drug eluting balloon (DEB). It is not clear why the DEB did not work, nor is there any attempt to explain the results. The DEB used in this study is a prototype, not commercially available, that had to be manipulated (shortened, ligated) to fit the rfat aorta. Therefore the chances of anyone reproducing this experiment are slim. The choice of PGG concentration is not explained, nor is the exposure time of 15 minutes. There is no proof of PGG binding to tissues (FeCl3 stain, PGG quantification) and thus, efficacy of delivery in this system is not clear. All experiments were done as attempts to treat with PGG immediately after elastase induction, which is not clinically relevant. Infusion of PGG after >50% AAA would have been more relevant. There were no differences in the mRNA levels of LOX and LOX1 which do not correspond to previous findings; no explanation is given for this also. The Legend to Figure 4 is not clear - where are these sections from? Finally, I would remove porcine form the title - the source of elastase is irrelevant.

6. PLOS authors have the option to publish the peer review history of their article (what does this mean?). If published, this will include your full peer review and any attached files.

Reviewer #1: No

Reviewer #2: No

---

## [Author Response · Author response to Decision Letter 0]

12 May 2020

All the comments and questions have been answered in the document uploadet: "Response to reviewers".

---

## [Decision Letter · Decision Letter 1]

27 May 2020

Intraluminal Infusion of Penta-Galloyl Glucose Reduces Abdominal Aortic Aneurysm Development in the Elastase Rat Model

PONE-D-20-06721R1

Dear Dr. Schack,

We are pleased to inform you that your manuscript has been judged scientifically suitable for publication and will be formally accepted for publication once it complies with all outstanding technical requirements.

With kind regards,

Michael Bader

Academic Editor

PLOS ONE

Additional Editor Comments (optional):

Reviewers' comments:

Reviewer's Responses to Questions

**Comments to the Author**

1. If the authors have adequately addressed your comments raised in a previous round of review and you feel that this manuscript is now acceptable for publication, you may indicate that here to bypass the “Comments to the Author” section, enter your conflict of interest statement in the “Confidential to Editor” section, and submit your "Accept" recommendation.

Reviewer #1: All comments have been addressed

Reviewer #2: All comments have been addressed

2. Is the manuscript technically sound, and do the data support the conclusions?

Reviewer #1: Yes

Reviewer #2: (No Response)

3. Has the statistical analysis been performed appropriately and rigorously? 

Reviewer #1: Yes

Reviewer #2: (No Response)

4. Have the authors made all data underlying the findings in their manuscript fully available?

Reviewer #1: Yes

Reviewer #2: (No Response)

5. Is the manuscript presented in an intelligible fashion and written in standard English?

Reviewer #1: Yes

Reviewer #2: (No Response)

6. Review Comments to the Author

Reviewer #1: (No Response)

Reviewer #2: (No Response)

7. PLOS authors have the option to publish the peer review history of their article (what does this mean?). If published, this will include your full peer review and any attached files.

Reviewer #1: No

Reviewer #2: No

---

## [Editor Report · Acceptance letter]

8 Jun 2020

PONE-D-20-06721R1 

Intraluminal Infusion of Penta-Galloyl Glucose Reduces Abdominal Aortic Aneurysm Development in the Elastase Rat Model 

Dear Dr. Schack:

I'm pleased to inform you that your manuscript has been deemed suitable for publication in PLOS ONE. Congratulations! Your manuscript is now with our production department. 

Kind regards, 

on behalf of

Prof. Michael Bader 

Academic Editor

PLOS ONE